# The Effects of 6-Month Vitamin D Supplementation during the Non-Surgical Treatment of Periodontitis in Vitamin-D-Deficient Patients: A Randomized Double-Blind Placebo-Controlled Study

**DOI:** 10.3390/nu12102940

**Published:** 2020-09-25

**Authors:** Marina Perić, Dominique Maiter, Etienne Cavalier, Jérôme F. Lasserre, Selena Toma

**Affiliations:** 1Department of Surgery, Medical, Molecular, and Critical Area, University of Pisa, 56126 Pisa, Italy; 2Department of Endocrinology and Nutrition, Cliniques Universitaires Saint Luc, Avenue Hippocrate 10, 1200 Brussels, Belgium; dominique.maiter@uclouvain.be; 3Department of Clinical Chemistry, CHU Sart-Tilman, University of Liège, 4000 Liège, Belgium; etienne.cavalier@chu.ulg.ac.be; 4Service de Parodontologie, Ecole de Médecine Dentaire—Cliniques Universitaires Saint Luc, Avenue Hippocrate 10, 1200 Brussels, Belgium; jerome.lasserre@uclouvain.be (J.F.L.); selena.toma@uclouvain.be (S.T.)

**Keywords:** periodontal disease, periodontitis, serum vitamin D, vitamin D deficiency, vitamin D supplementation, non-surgical periodontal treatment

## Abstract

Background: This study assessed the effects of weekly vitamin D (VD) supplementation on clinical and biological parameters after scaling and root planning (SRP) in the treatment of periodontitis and served to validate the VD dosage regimen. Methods: It was a monocentric, randomized, double-blind, placebo-controlled clinical trial with 6 months follow-up. Healthy Caucasian periodontitis patients presenting serum 25(OH) vitamin D3 below 30 ng/mL were randomly allocated to test group (SRP + VD 25,000 international units (IU)/week) or the control group (SRP + placebo). Results: A total of 59 patients were screened, 27 were included and 26 completed 3 months (M) and 21 completed 6M control. Test (*n* = 13) and control groups (*n* = 14) had similar 25(OH) vitamin D3 levels at baseline (17.6 ± 7.4 vs. 14.4 ± 5.2, respectively). After one month, there was a significant difference between groups (32.9 ± 5.2 vs. 16.1 ± 4.7), also seen at M3 and M6 (*t*-test, *p* < 0.001). Periodontal treatment was successful in both groups, since it resulted in a reduction of all measured clinical parameters at M3 and M6 (probing pocket depth (PPD), full mouth bleeding and plaque). However, the reduction in PPD was greater in the test group. Conclusions: In this short-term pilot study, no significant differences were observed between two groups. However, supplementation with VD tended to improve the treatment of periodontitis in patients with initial 25(OH) vitamin D3 < 30 ng/mL and proved safe and efficacious. NCT03162406.

## 1. Introduction

Periodontitis is a chronic inflammatory disease characterized by immunoinflammatory infiltrate in the deep compartments of the periodontium, leading to destruction of the tooth-supporting tissues, tooth mobility and eventually tooth loss. It is the result of a complex polymicrobial infection, which in susceptible individuals causes the perturbation of the homeostasis between the subgingival microbiota and the host defense [1,2]. The overall prevalence of severe periodontitis in 2010 was 11.2%, with an incidence rate of 701 cases per 100,000 person-year. In 2010, it was the sixth-most prevalent disease, with 743 million people affected worldwide [3]. Since the periodontal disease (PD) is among the most complex chronic, non-communicable diseases, there is an array of risk factors that need to be identified for every new patient [4]. Such risk factors as smoking, poorly controlled diabetes, obesity, osteoporosis, stress and low dietary calcium and vitamin D (25(OH)D3) have already been identified in 2013 by Genco and Borgnakke [5]. In this context, there is biological rationale to suspect that a vitamin D deficiency could negatively affect the periodontium, since vitamin D plays a crucial role in bone maintenance and immunity [6]. Vitamin D has been proposed to have anti-inflammatory properties that could be potentially appealing in the management of PD [7]. The inflammation, when it is fine-tuned, is one of the principal defense mechanisms of the body [8]. There is a controversy regarding the relationship between vitamin D and inflammation [9,10,11]. If inflammatory conditions decrease 25(OH)D concentrations and the body utilizes vitamin D to help heal and modulate inflammation, obtaining higher vitamin D serum concentrations might be of benefit in treating these disorders [9]. However, this hypothesis is not supported by some authors [12]. In addition, analysis of cross-sectional data of the third National Health and Nutrition Examination Survey (NHANES III) found that the serum vitamin D levels were significantly and inversely associated with bleeding on gingival probing in all age groups [13], and Dietrich et al. [14] reported a possible association between low serum vitamin D levels and PD. In a recent systematic review on vitamin D levels and PD, Pinto et al. [15] reported a significant association between low 25(OH)D levels and periodontal parameters in 65% of the cross-sectional studies analyzed. Still, among the observational longitudinal studies included, the authors found no proof that the PD progression could be attributed to lower serum 25(OH)D. Similar results were reported in another systematic review on vitamin D and PD [16]. In both reviews, however, the authors could not identify the interventional studies where only vitamin D supplementation was used as adjuvant in the treatment of PD [15,16].

Therefore, the present study wanted to test the hypothesis that vitamin D supplementation administered as an adjuvant to initial periodontal treatment and continued for 6 months in patients with generalized chronic periodontitis (GChP) and low serum vitamin D concentrations may lead to superior clinical results compared with the initial treatment alone. The primary objective of this study was to determine the effect of vitamin D on probing pocket depth (PPD). The secondary objective was to validate the applied vitamin D (VD) dosage regimen.

## 2. Materials and Methods

This was a randomized, placebo-controlled, double-blinded clinical trial with a 6-month follow up. The study was approved by the human subjects ethics board of the Cliniques universitaires Saint Luc, under the number 2017/06JAN/013, and by the Federal Agency for Medicines and Health Products (Agence Fédérale des Médicaments et des Produits de Santé (AFMPS)) under the Eudract number (medicinal clinical trial): 2016-005062-61, and was conducted in accordance with the Helsinki Declaration of 1975, as revised in 2013.

The study followed the CONSORT guidelines [17] for randomized clinical trials (Appendix A). The trial was performed according to the protocol Good Clinical Practices and the applicable regulatory requirements and registered at ClinicalTrials.gov under the Identifier: NCT03162406.

The participants were informed in detail of the nature and course of the study and signed an informed consent prior to their enrollment. Detailed medical, periodontal and dental history was obtained and subjects who fulfilled the inclusion criteria were invited to participate.

### 2.1. Inclusion Criteria

Eligible participants had to be 30 years or older Caucasians (defined as European and North African) in good general health and diagnosed with GChP based on the classification of the American Academy of Periodontology [18,19] Furthermore, they had to present a minimum of 15 teeth (excluding third molars and teeth with advanced decay indicated for extraction). The serum 25(OH) vitamin D3 concentration eligible for inclusion was set at <30 ng/mL.

Exclusion criteria comprised the following: pregnancy (tested with beta hCG—beta human chorionic gonadotropin test); breastfeeding; systemic diseases that could affect the progression of periodontitis (e.g., diabetes, immunological disorders, osteoporosis); scaling and root planning (SRP) in the previous 12 months; antibiotic therapy in the previous 6 months; long-term intake of anti-inflammatory medications; need for antibiotic pre-medication for routine dental therapy; any current ongoing immunological, neoplastic, endocrine, hematological, hepatic, renal, gastrointestinal, neurological, or psychiatric abnormalities or medical disease; subjects who used a UV light solarium or any type of vitamin D supplement within two months before the screening visit, or planned to travel outside European countries during the study; subjects under treatment with drugs that may interfere with vitamin D metabolism (e.g., phenobarbital, phenytoin, and glucocorticoids) and those with a past or current history of granulomatosis, especially sarcoidosis, urinary lithiasis, and osteomalacia; subjects who present a 25(OH)D concentration > 30 ng/mL, serum creatinine > 150 µmol/L and albumin corrected serum calcium > 2.65 mmol/L (corresponding to 10.6 mg/dL) at screening [20], and finally any sensitivity or allergy to any of the products that will be used in the study or a history of drug and/or alcohol abuse.

### 2.2. Clinical Protocol

Consort flow-chart is depicted in Appendix A. The participants were randomly allocated either to the group receiving the treatment under investigation (scaling and root planning (SRP) accompanied by administration of vitamin D) or to the control group receiving standard treatment (SRP in conjunction with placebo). The administration of vitamin D or placebo started one month prior to the beginning of the non-surgical periodontal treatment (M-1) and continued until the 6 months control (M6). Patients were asked to take one ampoule of 25,000 international units (IU) vitamin D3 per week (Table 1).

Scaling and root planing was performed in two sessions within 48 h, with the maximum duration of 1h30 per session, using Gracey curettes (Hu-Friedy, Chicago, IL, USA) and ultrasonic device (Newtron^®^, P5 XS B Led, Acteon, Satelec, France) under local anesthesia (Treatment Visit 2 (M0) and Treatment Visit 3 (M0)). Subsequently, participants were recalled at 1 month (M1), 3 months (M3) and six months (M6). At each session, oral hygiene instructions were repeated and reinforced. At M1, supragingival plaque deposits were removed, whereas, at M3 and M6, the residual periodontal pockets (≥5 mm) were re-instrumented and supragingival calculus was removed.

Blood samples were collected and analyzed at the Department of Clinical Laboratories, Cliniques universitaires Saint-Luc (screening visit, M0, M3 and M6).

The cobas e 602 automated electrochemiluminescent immunoassay (Roche, Basel, Switzerland) was used for 25(OH)D determination. In the lab, intra-, and inter-assay coefficient of variation are <6.8% and <13.1%, respectively. The assay measured both 25(OH)D2 and D3.

### 2.3. Randomization and Compliance

Random assignment of intervention was done after the participants had been assessed for eligibility and recruited, but before the beginning of the intervention (Appendix A). The randomization list was produced by the Department of Pharmacy of the hospital. The principal examiner (M.P.) and the participants were blind to the respective treatment arms until the end of the study. Allocation concealment was warranted by the use of identical vitamin D (D-Cure, SMB Laboratories, Belgium), or placebo ampoules, both produced by the company LABORATOIRES SMB S.A., Brussels, Belgium for the needs of the present study. After randomization, the two groups were followed in exactly the same way. All tested blood parameters are listed in the Appendix A, and respective timings in the Appendix A. The blood results were analyzed and monitored by the study endocrinologist (D.M.). Regular laboratory monitoring of blood levels of calcium, phosphorus, alkaline phosphatase, and 25(OH) vitamin D3 was performed to evaluate treatment safety and detect any excess of vitamin D. If hypercalcemia or kidney stone occurred, the subjects were withdrawn from the study. The compliance was assessed by asking the subjects if they adhered to the protocol and by measuring the 25(OH) vitamin D3 serum levels at M0, M3 and M6, that were systematically verified by the endocrinologist (D.M.). Participants were asked not to use any other vitamin D supplementation during the study period.

### 2.4. Calibration of the Examiner & Sample Size Calculation

Prior to initial treatment (M0), at 3 months (M3) and 6 months (M6), the following clinical parameters were assessed using a pressure-calibrated periodontal probe at a standardized contact pressure of 0.2 N (DB765R Aesculap, Tuttlingen, Germany): full mouth plaque score (FMPS) [21]; full mouth bleeding score (FMBS) [22]; probing pocket depth (PPD). All clinical measures were taken at 6 sites per tooth. At M1, only FMPS and FMBS were assessed. All clinical recordings, oral hygiene instructions and treatment sessions were performed by one single clinician (M.P.) blind to treatment allocation.

The calibration of the examiner was tested in 5 non-study patients with chronic periodontitis, each with a minimum of 10 teeth with at least one site with PPD ≥ 5 mm per tooth. The same protocol was repeated after 7 days. The calibration was accepted when ≥90% of the measures were within 1mm of agreement (Cohen’s Kappa Analyses). The sample size was calculated by taking the PPD as the main measure with standard deviation of 0.5 mm to detect a difference of 1 mm. It took 7 subjects per group to have a power of 0.90 and 6 per group for a power of 0.80 (alpha = 0.05). To compensate for a possible dropout, an additional 20% of subjects were deemed necessary.

### 2.5. Statistical Analysis

Bivariate analyses using a Student *t* test or χ^2^ test were performed to determine characteristics differing between the treatment group and control. Variables with normal distributions and equal variances across all levels of the independent variable parametric tests (*t*-test for independent samples) were used. In the case when the normal distribution in both groups was satisfied but equality of variance rejected, the Welch test was used.

For the values that did not satisfy normality in the independent *t*-test, the Mann–Whitney U test was performed. For the post-hoc test for the Kruskal–Wallis Mann–Whitney U test, Tukey’s test for highest significant difference was used (with correction of significance).

The results of the therapy were compared to identify the differences in the measured clinical parameters between baseline, 3 and 6 months after SRP for both groups. The pairwise *t*-test (based on differences) was used for paired measurements. The difference in the number of periodontal pockets ≥4 mm between baseline (M0) and the timepoint of interest was calculated for both groups and the intergroup comparison was made with Mann–Whitney U test.

Statistical significance of differences in categorical variables was determined using Fisher’s exact test (two dichotomous variables) or χ^2^-square test (for variables with more than two categories). In doing so, the φ correlation coefficient was used as a standardized measure of effect size. The level of significance was set at *p* < 0.05. In all analyses, two-tail tests of statistical significance were used. All of the statistical analysis was performed in SPSS 17.0 package (SPSS Inc., Chicago, IL, USA).

## 3. Results

The study was conducted at the Department of Periodontology, Cliniques universitaires Saint Luc, from April 2017 to October 2018. A total of 56 patients who had a Dutch Periodontal Screening Index (DPSI) > 2 were screened via blood measurement of 25(OH) vitamin D3 and a total of 27 were eligible to participate in the study. Out of 27 patients, 26 completed the 3M and 21 completed the 6M control. All recruited patients were Caucasians. The baseline characteristics are presented in Table 2. The two groups were comparable regarding all the assessed baseline parameters. There was an equal number of smokers in both groups. Moreover, the intention to treat and per protocol statistical analysis were conducted and yielded similar results.

### 3.1. Systemic Biomarkers

The systemic biomarkers were assessed at screening and after one and six months of VD supplementation in order to observe the impact of administered VD dosage on the systemic markers related to VD and calcium metabolism. There were no adverse events reported. All the measured systemic biomarkers remained in the reference range for both groups during whole duration of the study, which proved the safety dosage of the regimen (Table 3). When it comes to inter-group comparison, the only statistically significant difference was observed for parathyroid hormone (PTH) at M0 (*p* < 0.05).

### 3.2. 25(OH) Vitamin D3 Serum Levels

The patients in the test group received a total amount of 600,000 IU of VD in 6 months, with a weekly frequency of 25,000 IU. The two groups were statistically comparable at baseline regarding the serum 25(OH) vitamin D3 levels (*t*-test; *p* = 0.83), but, already after one month of supplementation, a statistically significant difference could be observed (*p* < 0.0001) (Figure 1). The difference between the test and the control group remained significant for the whole duration of the study (*p* < 0.0001 at M3; *p* < 0.001 at M6). The test group participants were compliant with the study protocol, which was tested with the serum 25(OH) vitamin D3 levels at M0, M3 and M6 (Figure 1). Already after one month of supplementation, all the patients belonging to the test group had their serum 25(OH) vitamin D3 levels ≥ 20 ng/mL. The test group patients reached the serum 25(OH) vitamin D3 levels > 30 ng/mL at M3 and M6, except for one patient who had serum 25(OH) vitamin D3 concentrations of 26 and 29 ng/mL, respectively.

When it comes to intra-group difference, test group exhibited a statistically highly significant difference at all timepoints (M0, M3 and M6 versus screening; paired *t*-test; *p* < 0.0001). The control group did not show a statistically significant difference at M0 and M3 versus screening (paired *t*-test; *p* = 0.13). However, at M6 there was a statistically significant difference (*p* < 0.01 versus screening). It is important to notice that this difference is possibly due to seasonal variations, since the 6M controls were done in the spring and summer period for 89% of patients.

The fluctuations in the levels of VD and PTH are inversely proportional, which is in accordance with the literature and the body’s regulatory mechanism and proves the safety of the treatment administered [20].

### 3.3. Clinical Parameters

Both treatment arms were efficient, since they resulted in a statistically significant intra-group reduction of all assessed clinical parameters for both groups at M6 (overall *p* < 0.01) (Table 4).

In the control group, the baseline number of PPD ≥ 4 mm of 49.64 (27.75–71.52) decreased to 20.45 (8.93–31.98) at 6 months control (*p* < 0.01). The test group followed the same pattern of diminution in the number of PPD ≥ 4mm, from the baseline 67.70 (48.44–86.96) to 24.80 (12.71–36.89) at M6 and this was statistically highly significant (*p* < 0.001).

The difference analysis for the number of PPD ≥ 4 mm revealed a greater reduction in number of PPD ≥ 4 mm in the test group at M6 (*p* = 0.072; Mann–Whitney U test) with respect to the control group (Table 4).

Even if this result is not statistically significant and is to be interpreted with caution, this difference could be clinically interesting and should not be neglected, especially considering the great intra-group reduction observed at M6 for test group (*p* < 0.001).

In terms of inter-group comparison for FMPS and FMBS, no statistically significant differences were observed.

## 4. Discussion

To the best of our knowledge, this is the first randomized controlled clinical trial (RCT) with vitamin D supplementation during non-surgical periodontal treatment in Caucasian patients with serum 25(OH)D levels below 30 ng/mL at the time of GChP diagnosis. Our data suggest that the weekly dosage of 25,000 IU of VD as adjunct to periodontal treatment is a safe and likely efficacious treatment method that could be associated with a higher degree of periodontal healing.

It is well known that bacteria are necessary, but not sufficient for periodontitis to develop [23]. In this regard, vitamin D and its deficiency have become an important source for investigation with promising evidence from case-control and cross-sectional studies that recognized vitamin D deficiency as a risk indicator for PD [15,16].

Accordingly, our study examined the effects of a VD supplementation of 25,000 IU per week. The proposed dosage regimen proved safe and efficacious for all the study participants, enabling the rapid increase in serum VD levels before the beginning of the periodontal treatment itself, and assuring stable serum VD levels during all the healing period of up to 6 months. All the measured biomarkers related to VD metabolism, as well as calcium levels, remained in the reference range throughout the study period and no adverse effects were observed. Consequently, this dosage regimen could be proposed as adjunct to periodontal treatment. At present, 25(OH) vitamin D concentration is still regarded as the best indicator of vitamin D status [24]. There is currently no standard definition or agreement as to “optimal” 25(OH)D levels, nor exists an agreement regarding 25(OH) vitamin D blood levels that can be qualified as “deficient”. As a result, there is a variety of cut-off points used in studies related to vitamin D [25,26]. Still, most clinical studies support the view that serum 25(OH)D levels of less than 20 ng/mL (50 nmol/L) indicate vitamin D deficiency, serum 25(OH)D levels below 30 ng/mL insufficiency, while levels between 30 and 60 ng/mL (75 and 150 nmol/L) represent normal values [10]. Another important issue, when interpreting and comparing the results of different studies, is the choice of vitamin D assay methodology and season or months in which blood samples were taken. In addition, the age range, ethnicity, sex, BMI and geographic latitude of the study population should be accounted for [26,27,28]. In this study, only Caucasian patients with vitamin D insufficiency were included, to exclude any potential racial bias. In order to diminish any possible seasonal bias, 89% of the patients (24 from 27) were included in the study and began with vitamin D/placebo supplementation during the autumn and winter seasons. In addition, a laboratory assay with a low percentage of intra-, and inter-assay coefficient of variation (<6.8% and <13.1%, respectively) was used. The reader should bear in mind that an increase in the VD serum levels in the control group can be observed. This is clearly visible at the last timepoint. Actually, the 6M control for 89% of patients was done in the spring and summer period with more sunny days in Belgium. Even if this cannot be concluded for test group, we speculate that a part of the increase could be attributed to seasonal variations in both groups. However, it is important to notice that a statistically significant inter-group difference remained at all timepoints.

Since vitamin D3 absorption from an oily supplement is not influenced by the presence or absence of a meal [29], the patients were instructed to take the ampoules depending on personal preferences. The actual dosage regimen (25,000 IU weekly; Table 2) was chosen to quickly increase the serum level of vitamin D before the SRP. In the study of De Niet et al. [30], it was proven that higher doses of vitamin D (50,000 IU) administered monthly increased faster serum concentrations than the daily doses (2000 IU), and the latter group reached a similar concentration to the monthly supplementation only at day 25 (25th day after the beginning of the treatment). Comparably, in the monthly group, the increase in 25(OH)_2_D3 versus baseline, was significant as early as day 2 (2nd day after the beginning of the treatment).

Our findings are in favor of available evidence from observational studies on VD and periodontitis [15,16] that support a perio-protective role of VD. There was a tendency towards a higher degree of periodontal healing in the test group, expressed through a greater reduction in the number of residual PPD ≥ 4 mm. Additionally, the test group showed a statistically highly significant intra-group diminution in the number of residual PPD ≥ 4 mm at 6M. Therefore, VD supplementation could have a beneficial impact on PD in promoting the healing after the non-surgical treatment of periodontitis. Nevertheless, this leaves only a likely hypothesis that needs to be confirmed by further studies.

The authors are well aware of the strengths and limitations of the current study. The principal limitation is the short follow-up period of 6 months and a small sample size. However, to the best of our knowledge, this is the first study on VD as the only adjuvant to SRP in a Caucasian population. Moreover, the supplementation started one month before the beginning of the treatment, assuring serum VD levels ≥ 20 ng/mL for all the test group patients at the time of the treatment. The second limitation is the lack of data on other biomarkers in serum, saliva and gingival crevicular fluid (GCF) (e.g., levels of pro-inflammatory cytokines like interleukin 1 beta (IL-1 β), interleukin 6 (IL-6), tumor necrosis factor alpha (TNFα), interferon gamma (IFNγ)) that have been related to PD [31,32,33]. It is worth mentioning here the alternative pathways of vitamin D activation and their function to gain a more thorough insight into the complex actions of this nutrient in the human body. Actually, a novel pathway of D3 metabolism was already introduced in 2012 and further tested by Slominski and coworkers [34,35], thus demonstrating the possibility of the activation pathway other than the classical one D3/D2->25(OH)D3/D2->1,25(OH)_2_D3/D2. It is initiated by mitochondrial enzyme CYP11A1, with the product profile showing organ/cell type specificity and is modified by enzyme CYP27B1 activity. However, its physiological importance is yet to be established.

The third limitation is the inclusion of smokers; that said, they were equally distributed in both groups. It is important to stress that the allocation concealment was assured using the identical placebo and that the compliance was assessed by a laboratory test at three timepoints, M0, M3 and M6, respectively. However, future well-conducted RCTs clarifying a clear link between the vitamin D and the onset and progression of PD are encouraged. Moreover, if vitamin D is to be established as a true risk factor for PD, then serum cutoff values of 25(OH)D related to PD should be identified as well.

## 5. Conclusions

In conclusions, our study is the first to report results of the effects of vitamin D as sole adjuvant to non-surgical periodontal treatment in a Caucasian population. The VD treatment regimen of 25,000 IU/weekly administered for 6 months proved safe and efficacious to restore normal 25(OH) vitamin D3 levels in patients with a periodontitis who underwent a non-surgical periodontal treatment. Moreover, there was a tendency towards a greater reduction in periodontal pockets ≥ 4 mm in the VD-treated group. The authors are well aware of the limitations related to a small sample size.

The research on the potential beneficial impact of vitamin supplementation in treating and preventing periodontal disease and its severe manifestations should be regarded as a marathon run. The same protocol of conducting a clinical trial as in a non-surgical periodontal therapy with antibiotic supplementation cannot be applied here, since the vitamin might not exert its effects as quickly. Consequently, the concept of 3- and 6-month controls after the initial treatment to verify its efficacy should be interpreted with extreme caution. On the other hand, a vitamin could prove its actions in the long run, a concept that needs to be tested in the clinical trials with well-designed and controlled follow-up periods. Therefore, future studies using the present data to influence the number of samples needed to detect differences are encouraged.

Vitamin D supplementation in conjunction with non-surgical periodontal treatment might be considered in the future as a treatment option in patients with initial serum VD below 30 ng/mL.

## Figures and Tables

**Figure 1 nutrients-12-02940-f001:**
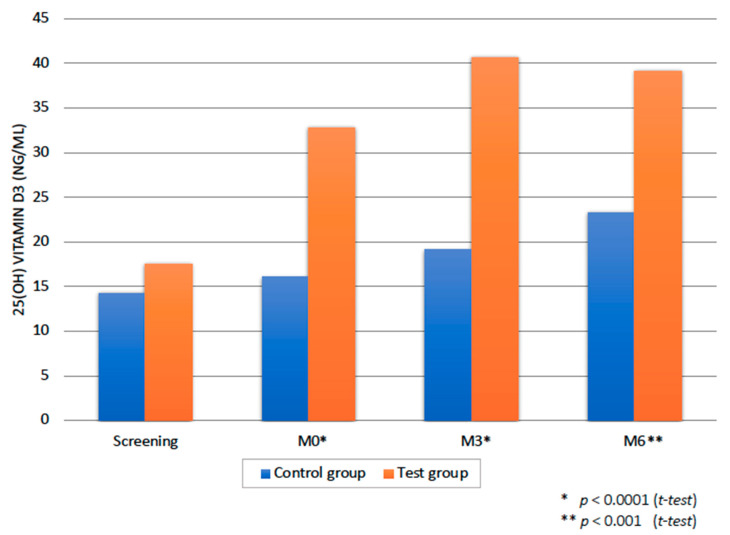
25(OH) vitamin D3 status over study timepoints. The p- values show inter-group difference at M0, M3 and M6 timepoints. * Significant effect of intensity.

**Table 1 nutrients-12-02940-t001:** Vitamin D/placebo administration.

Study Timepoints	Control Group	Test Group
Screening: Visit-1 (M-3 untill M-1)	X	X
Baseline Enrollment Randomization: Visit 1 (M-1)	1 ampoule of placebo per week	1 ampoule of 25,000 IU per week
Treatment Visit 2 (M0)		
Treatment Visit 3 (M0)		
Treatment Visit 4 (M1)		
Treatment Visit 5 (M3)		
Treatment Visit 6 (M6)		
TOTAL VITAMIN D in 6M	0 IU	600,000 IU

IU: international units.

**Table 2 nutrients-12-02940-t002:** Baseline characteristics.

Baseline Variable Mean ± SD	Control Group	Test Group
Number of patients (Intention to treat)	14	12
Number of patients (Per protocol analysis)	11	10
Age, years	46.18 ± 6.95	48.4 ± 9.08
Gender (female)	3	4
Smoking	4	4
Body mass index, (kg/m^2^)	26.61 ± 3.81	25.69 ± 5.82
Number of teeth	25.09 ± 2.63	24.7 ± 2.63
Creatinine (mg/dL)	0.85 ± 0.15	0.85 ± 0.16
Osteocalcin (ng/mL)	17.45 ± 4.35	18.27 ± 7.34
Alkaline phosphatase (U/L)	57.55 ± 14.32	72.9 ± 22.45
Bone alkaline phosphatase (U/L)	10.87 ± 3.74	13.16 ± 5.2
Total calcium (mmol/L)	2.38 ± 0.05	2.39 ± 0.03
High density lipoprotein (mg/dL)	56.18 ± 16.42	61.7 ± 19.29
Parathyroid hormone (pg/mL)	36.45 ± 11.51	38.9 ± 12.3
Neutrophils (10^3^/MicroL)	5.58 ± 2.12	5.07 ± 2.12
Lymphocytes (10^3^/MicroL)	2.84 ± 0.82	2.28 ± 0.85

**Table 3 nutrients-12-02940-t003:** Blood parameters.

Variable Mean ± Standard Deviation	Study Timepoints	Control Group	Test Group
Creatinine (mg/dL)			
	M0	0.84 ± 0.15	0.89 ± 0.14
	M6	0.77 ± 0.12	0.82 ± 0.15
Osteocalcin (ng/mL)			
	M0	18.78 ± 5.22	19.3 ± 5.43
	M6	17.28 ± 3.91	21.43 ± 9.86
Alkaline phosphatase (U/L)			
	M0	60.64 ± 11.36	77.3 ± 25.96
	M6	61.88 ± 12.79	74.22 ± 27.93
Total calcium (mmol/L)			
	M0	2.38 ± 0.09	2.45 ± 0.07
	M6	2.38 ± 0.06	2.42 ± 0.07
High density lipoprotein (mg/dL)			
	M0	55.36 ± 15.78	60.6 ± 18.02
	M6	56.66 ± 21.09	58.78 ± 17.95
Parathyroid hormone (pg/mL)			
	M0 *	41.36 ± 12.27	32.1 ± 5.17
	M6	36.33 ± 8.34	35.1 ± 4.51
Neutrophils (10^3^/MicroL)			
	M0	4.99 ± 1.82	4.62 ± 1.64
	M6	5.55 ± 2.31	4.46 ± 1.60
Lymphocytes (10^3^/MicroL)			
	M0	2.81 ± 0.84	2.35 ± 0.73
	M6	2.54 ± 0.77	2.33 ± 0.61

* *p* < 0.05.

**Table 4 nutrients-12-02940-t004:** Clinical parameters.

Variable	Study Timepoints	Control Group	Test Group	*p* between Groups
FMBS Mean ± SD	Baseline	0.41 ± 0.3	0.42 ± 0.3	NS
	M1	0.09 ± 0.06	0.07 ± 0.09	NS
	M3	0.18 ± 0.14	0.19 ± 0.05	NS
	M6	0.15 ± 0.11	0.12 ± 0.07	NS
	*p* intra-group (M6 vs. Baseline)	*p* < 0.01	*p* < 0.01	
FMPS Mean ± SD	Baseline	0.55 ± 0.37	0.45 ± 0.22	NS
	M1	0.23 ± 0.17	0.29 ± 0.19	NS
	M3	0.25 ± 0.19	0.28 ± 0.09	NS
	M6	0.13 ± 0.11	0.25 ± 0.17	NS
	*p* intra-group (M6 vs. Baseline)	*p* < 0.001	*p* < 0.01	
PPD Number (#) sites ≥ 4 mm Mean (95%CI)	Baseline	49.64 (27.75–71.52)	67.70 (48.44–86.96)	NS
	M3	24,73 (12.19–37.26)	36,20 (22.47–49.93)	NS
	M6	20.45 (8.93–31.98)	24.80 (12.71–36.89)	NS
	*p* intra-group (M6 vs. Baseline)	*p* < 0.01	*p* < 0.001	
Difference in PPD # sites ≥ 4 mm Mean (95%CI)	M3 vs. Baseline	24.91 (14.28–35.54)	31.50 (17.80–45.20)	NS
	M6 vs. Baseline	29.18 (16.76–41.60)	42.90 (29.82–55.98)	NS (*p* = 0.072)

#: Stays for number of sites. FMBS: full mouth bleeding score; FMPS: full mouth plaque score; PPD: probing pocket depth.

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
