# Peer review of "The Effects of 6-Month Vitamin D Supplementation during the Non-Surgical Treatment of Periodontitis in Vitamin-D-Deficient Patients: A Randomized Double-Blind Placebo-Controlled Study"

_nutrients, 2020, doi:10.3390/nu12102940_

Round 1

Reviewer 1 Report

 In a monocentric, randomized study, the authors recruited 27 periodontitis patients with serum 25(OH) vitamin D3 below 30ng/ml and randomly allocated them to test (SRP + VD 25,000 IU/week) or control group (SRP + placebo). After one month, there was a significant difference between groups (32.9 ± 5.2 vs. 16.1 ± 4.7), also seen at M3 and M6 (t-test, p < 0.001). Periodontal treatment was successful in both groups indicated by all measured clinical parameters at M3 and M6. You concluded that supplementation with 25,000IU/weekly vitamin D tended to improve treatment of periodontitis in patients with initial 25(OH) vitamin D3 < 30 ng/mL and proved safe and efficacious.

Main concern:

  1. 27 periodontitis patients with 25OHD <30ng/ml were included, and 26 patients completed the study, test (n = 13) and control groups (n = 14). But you only included 21 patients, (test 10) and (control 11), in your analysis (table 2). You should explain why you excluded 3 subjects (about 25% of your study population) in test group (one missed follow up and other two) and control group for analysis. What is subject number in table 3 and 4.
  2. The total study subject is small in both groups, plus about 40% subject is chronic smoker, as smoker is risk factor of periodontitis. I suggest the author should recruit more subjects and excluded the chronic smokers for final analysis if possible. Hopeful they can get P<0.05 not P= 0.072).
  3. Assay methodology for 25(OH)D3 with inter and intra assay variation should provide in your methods section.   Do you measure 25(OH)D [25(OH)D2 + 25(OH)D3] or 25(OH)D3? 
  4. Smoke can affect DBP and 25OHD levels (Jessil), for each smoker, how many PPD/years, dose the smoke pattern change during the 6 months periods in treat and control groups?.  

Minor concern:

  1. One patient with 25(OH)D3 of 26 ng/ml in 3 M and 29 ng/ml in 6 M. What is his or her screen 25OHD level and patient is a smoker? Or possible you should exclude this patient for your final analysis as patients still vitD insufficiency?
  2. In your discussion: At present 25(OH) vitamin D concentration is still regarded as the best indicator of vitamin D status in general population. The chronic inflammatory, periodontitis, may inhibit Vitamin D Binding Protein (DBP) synthesis (Youselzadeh) and total 25OHD may not truly reflect vitamin D nutrition status in chronic inflammatory situation.
  3. In your abstract: 27 were included and 26 completed the study. Test (n = 13) and control groups (n = 14). You may change to 21 were include in analysis Test (n=10) and control (n=11).
  4. In your discussion: serum 25(OH)D levels of less than 20 ng/mL (50 nmol/L) indicate vitamin D deficiency, serum 25(OH)D levels below 30 ng/mL insufficiency. If your choice 25OHD <20ng/ml, you may get better outcome (P<0.05 and not p= 0.072).
  5. Any BMI changes in 6 months in both groups?                                                                                                        P Youselzadeh,. Int J Endocrinol, vol. 2014; 2014. doi:10.1155/2014/981581 & N Jassil et al. Endocr Pract. 2017;23: 605-61. PMID: 28095044.

Author Response

To whom it may concern

I would like to express our gratefulness to the reviewers.

The questions raised helped us to critically evaluate our manuscript, to revise further some concepts and to clarify the most important aspects of it.

The authors are well aware of the strengths and intrinsic limitations of this study, especially those related to a small sample size.

The greatest value of this study lies in the fact that it is a pioneer study in vitamin D supplementation on Caucasian population affected by periodontitis with initial vitamin D levels below 30 ng/ml.

We strongly believe that the study design, with the beginning of the supplementation one month before the treatment, is not only scientifically valid but should be inserted in a standard of care for the Caucasian population affected by periodontitis with initial vitamin D levels below 30 ng/mL.

Moreover, the study is done during the treatment of this chronic inflammatory disease, which is the leading cause of tooth loss in the adult population and is related to a constant low-grade systemic inflammatory state, expressed by elevated concentrations of acute phase pro-inflammatory mediators leading to the inflammatory response. Accordingly, the results of our study opened the way to the new research in the field of periodontitis treatment using vitamin D supplementation of 25 000 IU per week.

All the measured biomarkers related to vitamin D metabolism, as well as calcium levels, remained in the reference range throughout the study period and no adverse effects were observed. Consequently, this dosage regimen could be proposed as adjunct to periodontal treatment.

Thank you for your consideration of this manuscript.

Sincerely,

Marina Peric

Reviewer 2 Report

1. In the author's power calculation, the authors fail to adjust for multiple comparisons. Currently, the power calculation is done for only one statistical test. The authors on the other hand conduct many statistical tests. The power calculation part of the manuscript should be adjusted to reflect these multiple comparisons, or the authors should make it clear as to what specific comparison the study is powered to test. 2. The power calculation part seems to suggest that participants PPD measurements will have a standard deviation of 0.5mm. Table 4 suggest that this is far from the truth. The power calculation part seems to be based on numbers from the measurement tool, although the authors are not making inference on the measurement tools, rather between patients in different groups. The assumed standard deviation should be based on what the average expected deviance of PPD is in the patient population. 3. Table 1’s formatting needs some work. I understand what they were wanting to do here although the formatting makes it hard to understand. 4. In section 3.2, what was the p-value for the comparison of the two groups at baseline? The authors should report that in the manuscript. The authors say that they were comparable. Is that statistically comparable? Clinically comparable? Please be more specific. 5. Do the authors have a justification as to why the control groups VIT D3 levels went up from screening to M6? This should be discussed. If the control group showed increases, obviously not due to the supplementation, then could part of the increase that was seen in the treatment group been due to something besides the supplementation? 6. In section 3.3 the authors should report the reduction (M6 vs Baseline) of PPD>=4mm in the treatment and control groups. It is in table 4, although it would be nice to see inline. 7. A statement should be included in 3.3 that says that no other differences were seen between groups. Also, the statement “The difference analysis for the number of PPD ≥ 4mm revealed a greater reduction in number..” goes on to report a p-value>0.05 which the authors previously state is their cut off for statistical significance. I would at the least add a few words in that sentence letting the reader know that this difference is clinically interesting although not statistically significant (p-value=0.072). 8. In the tables and figures the authors switch between using a “.” As a decimal place and a “,”. Please use one consistently. 9. Figure 1 is not acceptable in its current format. Microsoft Word’s red lines under words should be removed at the least. Also, it is this reviewer’s opinion that 3d column charts distract from the data being presented. A regular 2d bar chart with different colors for the control and test groups would be clearer. Also, for figure 1, please see previous comment about “.” And “,”. 10. Misspelled “begining" should be “beginning” . Other misspellings: clafifying around line 275. 11. The power analysis section is misleading and suggests that only a few subjects are needed to test differences in # of PPD sites between two groups. The used estimate for SD (0.05mm) is not used correctly. The power analysis should be done on the number of PPD sites>=4mm since that is the main outcome of interest here. That would mean that the authors should have estimated what difference would have been clinically interesting to see and what is the estimated standard deviation of differences. From what I can see from the 95% CI’s it looks like the standard deviation for differences from M6 to baseline may be around 6 or 7. This change in the estimates and the correct use of an outcome for the power analysis would likely recommend many more patients then were included in this study. While I applaud the authors for conducting, possibly, the first RCT on this subject, it appears that a lack of baseline data may have mislead them into the number of patients that they needed to sufficiently study this phenomenon and detect differences. 12. I think the conclusion section is well written although I think there should possibly be more emphasis put on the limitations of sample size and that future studies should use these data to influence the number of samples that are needed to detect difference.

Author Response

(The authors gave the same response as above.)

Round 2

Reviewer 2 Report

Thank you for making the improvements. The manuscript is well written and I applaud your group on this work. 

Author Response

I would like to express my sincere gratitude to the reviewer on behalf of our research team for the detailed work and most pertinent comments. Actually, the peer-review process allowed us to accentuate the strengths and discuss the limitations of the manuscript, bringing into light all the possible pathways for future trials related to vitamin D in the field of periodontology.

Marina Peric